# *Gambierdiscus* and Its Associated Toxins: A Minireview

**DOI:** 10.3390/toxins14070485

**Published:** 2022-07-14

**Authors:** Da-Zhi Wang, Ye-Hong Xin, Ming-Hua Wang

**Affiliations:** State Key Laboratory of Marine Environmental Science, College of the Environment and Ecology, Xiamen University, Xiamen 361105, China; xinyehong1995@163.com (Y.-H.X.); mhwang45@xmu.edu.cn (M.-H.W.)

**Keywords:** *Gambierdiscus*, ciguatoxins, maitotoxin, ciguatera fish poisoning

## Abstract

*Gambierdiscus* is a dinoflagellate genus widely distributed throughout tropical and subtropical regions. Some members of this genus can produce a group of potent polycyclic polyether neurotoxins responsible for ciguatera fish poisoning (CFP), one of the most significant food-borne illnesses associated with fish consumption. Ciguatoxins and maitotoxins, the two major toxins produced by *Gambierdiscus*, act on voltage-gated channels and TRPA1 receptors, consequently leading to poisoning and even death in both humans and animals. Over the past few decades, the occurrence and geographic distribution of CFP have undergone a significant expansion due to intensive anthropogenic activities and global climate change, which results in more human illness, a greater public health impact, and larger economic losses. The global spread of CFP has led to *Gambierdiscus* and its toxins being considered an environmental and human health concern worldwide. In this review, we seek to provide an overview of recent advances in the field of *Gambierdiscus* and its associated toxins based on the existing literature combined with re-analyses of current data. The taxonomy, phylogenetics, geographic distribution, environmental regulation, toxin detection method, toxin biosynthesis, and pharmacology and toxicology of *Gambierdiscus* are summarized and discussed. We also highlight future perspectives on *Gambierdiscus* and its associated toxins.

## 1. Introduction

*Gambierdiscus* is a marine benthic dinoflagellate genus widely distributed throughout the world’s tropical and subtropical regions [1]. *Gambierdiscus* species are autotrophic benthic microalgae living on macrophytes, corals, and sand grains [2,3]. Members of this genus are notorious for producing a group of potent polycyclic polyether neurotoxins that can specifically activate voltage-gated sodium channels (Nav) [4] and inhibit neuronal potassium channels (Kv) [5], which increases neuronal excitability and, consequently, results in human disease. Ciguatoxins (CTXs) and maitotoxins (MTXs) are the two major toxins produced by *Gambierdiscus* [6,7]. CTXs can accumulate in benthic-feeding organisms and can subsequently bioconcentrate in top-predator reef fishes through transfer along the food chain [8]. When humans ingest CTX-contaminated fish or shellfish, they can develop a type of food poisoning known as ciguatera fish poisoning (CFP; or just ciguatera) [6,9,10,11]. Although the symptoms of CFP are nonspecific, they primarily manifest in the digestive, joint, muscle, cardiovascular, and nervous systems [12]. It was estimated that ciguatera affects 50,000–500,000 people worldwide every year [13]. Although this disease has existed for centuries, its diagnosis, prevention, treatment, and management still present major challenges [14,15].

Over the past few decades, substantial research effort has been devoted to *Gambierdiscus* and its toxins [16,17,18], and great advancements have been made in deciphering its taxonomy, phylogenetics, geographic distribution, toxin detection method, biosynthesis, toxicology, and pharmacology [19,20,21,22]. Notably, the occurrence and geographic distribution of CFP have undergone a considerable expansion due to intensive anthropogenic activities and global climate change, rendering it a worldwide concern [23]. The clinical features, pathophysiological basis [15], distribution [24], and detection method [25] of *Gambierdiscus*-induced CFP have been reviewed, but a systematic review of *Gambierdiscus* and its associated toxins is still lacking. In this review, we seek to fill the above knowledge gaps and refresh our understanding of *Gambierdiscus* and its associated toxins. We summarize the progress concerning the taxonomy, phylogenetics, geographic distribution, role of environmental factors, toxin detection method, toxin biosynthesis, pharmacology, and toxicology of *Gambierdiscus* and discuss the future perspectives for *Gambierdiscus* and its associated toxins.

## 2. Taxonomy and Phylogenetics of *Gambierdiscus*

*Gambierdiscus* (Gonyaulacales, Dinophyceae) species are armored, benthic dinoflagellates predominantly living in coral reef ecosystems attached through mucous filaments to the surfaces of macroalgae, seagrasses, and other substrata [3,26]. The morphology of *Gambierdiscus* has been extensively studied since 1978. Cells are large-sized (diameter 42 to 140 µm) [27], with strong anteroposterior compression and an ascending cingulum with a recurved distal end, and contain several yellow to brown chloroplasts [28]. *Gambierdiscus* species are traditionally identified based on subtle differences in their thecal plate morphology as observed under light microscopy and scanning electron microscopy [28,29]. According to the Kofoidian nomenclature of dinoflagellate thecal plate series for armored species, the theca is divided into various plates, such as apical pore (Po), apicals (′), precingulars (″), postcingulars (′′′), and antapicals (′′′′), among others. For *Gambierdiscus*, the plate formula is Po, 3′, 7″, 6c, 6-8s, 5′′′, 1p, 2′′′′ (Figure 1) [30,31]. Litaker et al. used dichotomous trees to distinguish 10 *Gambierdiscus* species based on their cell size, shape, and plate structure [30]. To date, a total of 18 *Gambierdiscus* species have been identified, including *G. australes*, *G. balechii*, *G. belizeanus*, *G. caribaeus*, *G. carolinianus*, *G. carpenteri*, *G. cheloniae*, *G. excentricus*, *G. honu*, *G. jejuensis*, *G. lapillus*, *G. pacificus*, *G. lewisii*, *G. holmesii, G. polynesiensis*, *G. scabrosus*, *G. silvae*, and *G. toxicus* [32], while some have yet to be classified.

However, it remains challenging to distinguish different *Gambierdiscus* species based on morphology alone because of their high similarities (Figure 1). Furthermore, the morphological approach alone does not properly allow an accurate identification at the species level and should be combined with molecular analysis. The sequencing of ribosomal (r) RNA-encoding DNA, including SSU rRNA, D1–D3 LSU rRNA, and D8–D10 rRNA genes, has been employed for the identification of *Gambierdiscus* species since the 1990s [33,40,41]. *Gambierdiscus* species show similar phylogenetic relationships in phylogenetic trees constructed based on different rRNA gene regions (Figure 2). The SSU region exhibits a lower substitution rate among species and a higher substitution rate among genera, while the LSU region displays the opposite trend (Figure 2). This is consistent with a study that showed that, in some dinoflagellates, the D1–D6 regions of the LSU rRNA gene have a substitution rate 4–8% faster than that for the whole SSU rRNA gene sequence [42]. The D8–D10 rRNA gene is the most commonly reported of the three sequences in the NCBI database, suggesting that the D8–D10 rRNA gene is the most suitable for identifying *Gambierdiscus* species. Notably, the classification of *G. carpenteri* in the phylogenetic trees constructed using D8–D10 rDNA and SSU rDNA is not uniform (marked in black in Figure 2), indicating that care must be taken when using these sequences to identify this species.

It should be pointed out that some species from the genus *Fukuyoa*, other important benthic algae resulting in CFPs [43,44], were classified in *Gambierdiscus* before 2015 due to their high morphological similarity and were often studied and discussed alongside *Gambierdiscus* [44,45,46]. However, recent studies showed that these two genera not only differ in morphology (species in *Gambierdiscus* with lenticular shape, and species in *Fukuyoa* with globular shape) but also belong to different branches based on the sequencing results of LSU (large subunit) and SSU (small subunit) ribosomal DNA [47]. Therefore, the following discussion is focused on *Gambierdiscus*, considering its more diverse and broader distribution than *Fukuyoa* in the ocean, as well as its ecological and human health concern.

## 3. Geographic Distribution and Role of Environmental Factors

Traditionally, *Gambierdiscus* species were viewed as pantropical organisms and widely distributed throughout tropical and subtropical regions of the world [28,30,42], especially in coastal areas of the Caribbean Sea [48,49], Indian Ocean [50], and Pacific Ocean [51] (Figure 3). However, the presence of *Gambierdiscus* has also been reported in temperate waters, including the nearshore waters of Australia [20], Japan [52], and the Mediterranean [28].

To further understand the global distribution of *Gambierdiscus*, we constructed a phylogenetic evolutionary tree using the D8–D10 LSU and SSU rRNA regions combined with the distribution information obtained from the Ocean Biodiversity Information System and the IOC Harmful Algal Bloom Programme [44,53,54]. In a phylogenetic analysis-based study, Litaker et al. (2010) reported that five *Gambierdiscus* species are endemic to the Atlantic (including the Caribbean/West Indies and the Gulf of Mexico), five are endemic to the tropical Pacific, and that two (*G. carpenteri* and *G. caribaeus*) are globally distributed. However, *G. belizeanus*, an Atlantic species following Litaker et al. [55], was later reported in the Central Pacific by Xu et al. [56], suggesting that some *Gambierdiscus* species might have been transferred via modern shipping activities [57]. Rodríguez et al. suggested that the Canary Islands (North-East Atlantic) could represent ancient settlement sites for *Gambierdiscus* as suggested by the high species diversity in the area [58], however, there is still not enough evidence to prove this hypothesis. The dispersal–vicariance analysis performed in this study using RASP (Figure 3) [59] shows that some widely distributed species, such as *G. carpenteri* and *G. caribaeus*, are scattered in different clades of the tree.

The growth and proliferation of *Gambierdiscus* cells are influenced by diverse environmental factors, among which temperature, salinity, and irradiance are thought to be key [60,61,62,63,64,65]. Laboratory studies have shown that the capacity for environmental adaptation of *Gambierdiscus* shows marked variation among species and even within species [66]. For example, *G. belizeanus*, *G. caribaeus*, *G. carpenteri*, and *G. pacificus* generally exhibit a wider range of tolerance to environmental conditions [66], consistent with their broad geographic distribution (Figure 3). In contrast, *G. silvae*, *G. australes, G. scabrosus*, and *G. jejuensis* showed a narrow range of tolerance to temperature, salinity, or irradiance [33,66,67]. *Gambierdiscus* achieves maximum cell growth in the temperature range of 25–31 °C. Both field observations and laboratory experiments have shown that some *Gambierdiscus* species, such as *G. carolinianus* and *G. caribaeus*, can tolerate low-temperature environments (<20 °C) [62,68]. Unlike most strains of *G. caribaeus* and *G. carpenteri*, which can survive at temperatures ranging from 33.6 and 35.4 °C, *G. jejuensis* strains cannot tolerate water temperatures above 30 °C [33]. *G. jejuensis* and *G. carpenteri* share the same clade in the phylogenic tree (marked in red lines in Figure 2), indicating that evolutionarily similar species can have differential capabilities for environmental adaptation. Several pan-genome analyses have been undertaken to examine the adaptation to the environment and the evolution of organisms from different habitats [69,70,71], however, no publicly available genome database currently exists for any *Gambierdiscus* species. If the warmer waters can meet their growth requirements, there may be a positive correlation between temperature and *Gambierdiscus* abundance [72]. Over the past 10 years, the number of *Gambierdiscus* occurrence areas and the number of CFP cases reported from tropical and subtropical regions have increased due to ocean warming [73,74]. Irrespective of whether these distributions in temperate habitats are temporary or permanent, the expanding distribution of some *Gambierdiscus* species is linked to the risk of broadening the endemic range of CFP occurrence.

Salinity is another important environmental factor affecting the distribution and growth of *Gambierdiscus*. The global distribution of *Gambierdiscus* species is linked to their capacity for adaptation to varying salinity [62]. Most species achieve their maximum growth in the salinity range of 25–35 [60,61,64,65]; however, some species, such as *G. caribaeus*, can adapt to a wider salinity range (15–40) [67]. 

Members of the *Gambierdiscus* genus depend on light to produce energy for their physiological activities [75]. As a typical benthic genus, *Gambierdiscus* achieves optimum growth under low irradiance conditions (49–231 μmol photons m^−2^ s^−1^) in the laboratory [62,75]. In the natural environment, deeper water layers may have weaker light conditions. But in the study of Xv et al., higher abundances of *Gambierdiscus* species were observed in shallower waters than in deeper waters, however, this is not yet certain to be related to the difference in light intensity [56]. Another study showed that there is no significant relationship between depth and *Gambierdiscus* [76]. Notably, not all *Gambierdiscus* are affected by photoinhibition (e.g., *G. silvae*) [75]. The strategies used by different *Gambierdiscus* species to adapt to different light intensities, especially low light intensity, remain to be explored.

Interestingly, nutrients are key factors affecting phytoplankton growth, but there are still no studies demonstrating significant effects of their concentration, types, and ratios on the cell growth of *Gambierdiscus* [68,77]. In addition to temperature, salinity, and irradiance, grazing pressure is also an important factor regulating *Gambierdiscus* abundance in the field. As an epiphytic dinoflagellate, *Gambierdiscus* cells are first consumed by herbivorous fish grazing on macroalgae that host them, then these cells are further transferred to carnivorous fish, such as grouper or snapper, through the trophic chain, which affects cell abundance. Meanwhile, ciguatoxins produced by *Gambierdiscus* are accumulated in these fishes, especially in fatty tissues, liver, viscera, and eggs, which provides new insights to address the prevalence of toxicity in the food web [76]. Overall, these studies indicated that the effect of environmental factors on *Gambierdiscus* is complicated, and more efforts should be devoted to interactions between different *Gambierdiscus* species and environmental factors to enhance our understanding of *Gambierdiscus* in future marine environments under the frame of global climate change.

## 4. *Gambierdiscus*-Associated Toxins

Many species in the genus *Gambierdiscus* can produce CTXs and/or MTXs, as well as their analogs (Table 1) [28,31]. These toxins are responsible for cases of CFP worldwide and pose a potential risk to human health. To date, more than 30 CTX congeners have been identified. They are classified into CTX3C, Caribbean Sea CTXs (C-CTXs) [78], Pacific Ocean CTXs (P-CTXs/CTX4A) [12], and Indian Ocean CTXs (I-CTXs) [79] based on the make-up of the structural backbone of each molecule [80]. CTXs are lipophilic, ladder-shaped polyethers with 13–14 cyclic consecutively connected rings (Figure 4) [81] and have similar structures to yessotoxins and brevetoxins.

Although some MTXs display higher toxicity than CTXs, their roles in CFP are still unknown [10], likely because MTXs have lower oral potency and greater water solubility than CTXs, the latter of which renders it difficult for MTXs to accumulate in fish and invertebrates [91]. Although MTX was first isolated from surgeonfish (*Ctenochaetus striatus*, “maito” in Tahiti) [92], it was then found to be produced by *G. polynesiensis*, *G. australes*, *G. belizeanus,* and *G. excentricus*. To date, six congeners of MTX have been identified, and most have been structurally elucidated [93,94]. Using ChemDraw (v20), we predicted and compared the 2D and 3D structures of *Gambierdiscus*-associated toxins (Figure 4). Like CTXs, MTXs are also polyether compounds, but the molecular masses of different MTXs vary greatly. Most MTXs are larger than CTXs, but part of their structure is similar to that of CTXs (Figure 4) [95].

In addition to CTXs and MTXs, members of the genus *Gambierdiscus* also produce gambieric acids (GAs), gambieroxide, gambierol, and gambierones [24,96]. GAs are polycyclic ethers first isolated from indoor-cultured *G. toxicus* [88]. They have since been detected, together with CTX homologs, in shark tissues [97]. Four types of GA have been identified, named GA A–D. GAs have antifungal activities, especially against filamentous fungi [88]. In addition to defense functions, GAs are also thought to have a role in the regulation of *G. toxicus* growth [24]. Gambieroxide is a type of epoxy polyether compound first isolated from *G. toxicus* strain GTP2 from Tahiti (French Polynesia). Gambieroxide has putatively been detected in *G. australes* strains from Menorca and Mallorca (Balearic Islands, Spain) [83]. The structure of gambieroxide is very similar to that of yessotoxin, containing 12 contiguous *trans*-fused rings comprising 6–8 carbons, one sulfate ester group, one epoxide, and two olefins in their side chains [89]. Gambierol is a ladder-shaped, *trans*-fused, octacyclic ring system with 43 carbons and high lipophilicity [98]. This toxin can bind to voltage-gated potassium channels in several tissues, thereby inhibiting K^+^ currents [99,100]. Gambierones are also polyketide compounds isolated from *G. belizeanus* (strain CCMP401). They contain a noncyclic polyether core with a complex side chain at both extremes. Gambierones purified from *G. cheloniae* CAWD232 exhibit substantially lower toxicity than P-CTX1B, indicating that gambierones are unlikely to be hazardous to human health [86]. Overall, these biologically active substances render *Gambierdiscus* potentially suitable for application in the field of biomedicine.

## 5. Toxin Detection Methods

Identifying toxic species and/or strains is an efficient strategy for the prevention of CFP at the source. However, the ability of different *Gambierdiscus* species to produce toxins cannot be predicted based on rDNA. Although different *Gambierdiscus* species can produce the same toxins (Table 1), suggesting that members of this genus may have acquired the ability to produce toxins early in their evolution, even different strains of the same species can display widely varying capabilities for toxin production [44,101,102,103]. Accordingly, there is an urgent need to develop *in situ* methods that can measure the toxicity of *Gambierdiscus* species. Techniques involving fluorescence in situ hybridization (FISH) probes and recombinase polymerase amplification have been developed and applied in the field for the in situ detection of *Gambierdiscus* spp. as well as other species that cause CFP [104,105,106]. FISH probes allow the *in situ* counting of *Gambierdiscus*. Recombinase polymerase amplification can detect the presence of even a single *Gambierdiscus* cell and shows high species specificity [107]. Notably, probe design in these methods is still based on rDNA sequences [104]. Toxin gene-based species detection techniques have been widely used to detect pathogenic bacteria and have achieved good results [108,109,110]. Similar methods, based on toxin-related genes, need to be also widely applied to *Gambierdiscus*.

CTXs are colorless, odorless, and thermally stable and cannot be destroyed by cooking or freezing [111]. Although the concentrations of toxins in *Gambierdiscus* and fish samples are low, they have a high toxicity. Because diverse toxin analogs exist [18], detecting these toxins in environmental samples has been challenging. Over the past few decades, various analytical methods, including biological, chemical, and immunological methods, have been introduced to detect and characterize CTXs to support fish product monitoring and protect human health. Bioassay methods that use the mongoose, mouse, cat [112], brine shrimp, mosquito, chicken, and dipteran larvae have been developed to detect CTXs in fish [113]. However, due to ethical and cost concerns, it is unlikely that large animals will continue to be used for CTX detection. The use of two of the above-mentioned test animals, brine shrimp and mosquitoes, has also been banned [113]; brine shrimp cannot effectively detect toxins contributing to CFP and it is unsuitable to cultivate mosquitoes in the laboratory [114]. The mouse bioassay is the only animal assay that continues to be applied [115]. Cell-based assays can help detect CTXs [116]. Regardless of the shortcomings of this method (Table 2), it is often employed in combination with other CTX detection methods [80,117,118]. Immunoassays such as radioimmunoassays [119], enzyme immunoassays [120], antibody-based immunoassays [121,122], membrane immunobead assays [123], enzyme-linked immunosorbent assays [124], and capillary electrophoresis-based immunoassays [125] are also utilized to detect CTXs. However, these immunoassays all have their limitations, and they have not been widely applied, even though some have been commercialized (Table 1) [126,127,128]. Liquid chromatography with tandem mass spectrometry (LC-MS/MS) is the most widely used method for the accurate identification of CTX types in toxin-contaminated samples [129,130,131]. However, this method is also limited by the lack of toxin standards and the impossibility of field application [25]. Electrochemical immunosensors have been designed to measure in situ CTXs in recent years, as they can be integrated into compact analytical devices such as smartphones [118]. It is expected that this method will be applied with good results to the detection of CTXs in the future. Although many methods have been developed, none are widely used in detecting and identifying CTXs in fish and fishery products because of cost and efficiency concerns and the complexity of the procedures involved [25].

## 6. Toxin Biosynthesis

Several studies have shown that environmental factors affect toxin production and accumulation in *Gambierdiscus*. By comparing the growth rate and toxin production of *G.*
*carpenteri* under different temperatures, light, and salinity, Vacarizas et al. found that cells produce more toxins during the slowest growth rate at a certain range of environmental conditions, and the highest cellular toxin content recorded was 7.48 ± 0.49 pg Pbtx eq/cell at culture conditions of 25 °C, 100 μmol photons m^−2^ s^−1^, and salinity of 26 [134]. The asynchrony between the abundance and toxicity of *Gambierdiscus* was also observed in a field study [135]. These results suggest that *Gambierdiscus* allocates more energy to growth and division than to toxins synthesis under suitable conditions. Although temperature affects cell growth and proliferation of *Gambierdiscus*, it is not regarded as an essential factor in the regulation of toxin production of *Gambierdiscus* spp. [49]. Longo et al. compared the levels of CTXs and its congeners in *G. polynesiensis* under different pH, N:P ratios, and nitrogen sources, and they found that more oxidized P-CTX analogs with higher potential toxicity are produced under low pH conditions [77]. These studies provide us with a snapshot of toxin production in *Gambierdiscus*.

Although the cellular processes underlying the biosynthesis of CTXs and MTXs remain unclear, some studies provide hints as to some of the cellular processes involved in the biosynthesis of the two toxins. Both CTXs and MTXs are polyether toxins. The synthesis of polyketides mediated by polyketide synthase (PKS) is regarded as essential for the biosynthesis of both toxin types [136]. Typically, PKS builds carbon chains in a manner similar to fatty acid synthase (FAS), where the starting substrate, usually acetyl-coenzyme A (acetyl CoA), is joined to malonyl CoA through a series of successive Claisen ester condensation reactions. The core structure of PKS consists of ketosynthase (KS), acyltransferase (AT), and acyl carrier protein (ACP) domains. Other domains, such as the dehydratase (DH), ketoreductase (KR), and enoylreductase (ER) domains that serve to modify the condensed acyl-units, are not essential for PKS function but are important for the synthesis of mature toxins. Another domain often found in PKS is thioesterase (TE), which is proposed to release polyketide compounds from megasynthase [137]. Epoxide-opening cascade reactions are also postulated to be involved in toxin biosynthesis, but this possibility remains to be confirmed [138].

Recently, transcriptome-based studies have been undertaken to investigate the PKSs in *Gambierdiscus* [138,139,140]. A comprehensive transcriptomic analysis of two gonyaulacaleaen and MTX-producing *Gambierdiscus* species, *G. australes* and *G. belizeanus*, identified genes putatively involved in the biosynthesis of polyether ladder compounds. Among these genes, 306 were found to be involved in polyketide biosynthesis, including 192 encoding ketoacyl synthases, and formed five unique phylogenetic clusters [139]. Interestingly, two clusters were unique to these maitotoxin-producing species, suggesting that they might be associated with MTX biosynthesis [140]. Furthermore, a putative biosynthetic pathway for MTX-1 is proposed, in which the carbon backbone is synthesized via polyketide biosynthesis followed by epoxidation, polyepoxide cyclization, and sulfonation carried out by PKSs, epoxidases, epoxide hydrolases, and sulfotransferases, respectively [139]. A recent comparative transcriptomic study of a CTX-producing strain and a non-CTX-producing strain of *G. balechii* identifies 52 PKS genes that were upregulated in the CTX-producing strain, including transcripts encoding both single-domain and multi-domain PKSs, suggesting that PKSs are likely to be involved in polyketide synthesis and also potentially CTX synthesis in *G. balechii* [103]. Collectively, these studies laid the foundation for elucidating the mechanisms involved in the biosynthesis of CTXs and MTXs and provided candidate biomarkers for the identification of toxin-producing *Gambierdiscus* species.

Dimethylsulfoniopropionate (DMSP), an organosulfur compound and zwitterionic metabolite, has been identified in many marine algal species [141,142,143]. DMSP is involved in numerous important biological processes, including cryoprotection [144], the scavenging of reactive oxygen species [145], and osmoregulation [145]. *Gambierdiscus* species are important producers of DMSP in the ocean and present a potential connection between DMSP and toxin production, given that DMSP has been proposed to serve as a signaling molecule for toxin synthesis [141]. However, this supposition requires further verification.

Chemical methods are also used to artificially synthesize CTXs and MTXs for potential biological applications as well as further studies of these toxins. Hirama et al. first reported the total synthesis of a CTX (CTX3C) [81]. Since then, a variety of strategies have been developed for the chemical synthesis of CTXs, including the most toxic ones and 51-hydroxyCTX3C [81]. MTXs are thought to be the largest and most toxic secondary metabolites isolated and identified to date; however, only fragments of this toxin have been synthesized. In summary, the chemical synthesis of CTXs and/or MTXs may help shed light on the mechanisms involved in their biosynthesis in *Gambierdiscus*. Additionally, synthetic toxins can be employed in toxicological and pharmacological studies.

## 7. Toxicology and Pharmacology

All CTXs can activate voltage-gated sodium channels and block potassium channels [146,147,148]. They can also transverse the blood–brain barrier, causing neurologic symptoms in both the central and peripheral nervous systems, as well as affecting the cardiovascular system [15,149]. The major symptoms of ciguatoxin poisoning occur within 1–3 h of ingesting toxin-contaminated fish and manifest as vomiting, diarrhea, numbness in the extremities, numbness in the mouth and lips, reversal of hot and cold sensations, and muscle and joint aches [150]. Moreover, 20% of people affected may develop chronic ciguatera poisoning, and chronic weakness may last for years [151]. Despite some efforts, there is still no antidote for any natural marine toxin [152]. Transcriptome-level studies have been devoted to studying the effects of CTXs on mice, both in vitro and in vivo, as well as on the whole blood of patients [153,154,155]. This will contribute to the understanding of the mechanisms associated with the symptoms and the responses of organisms to these toxins. Indeed, dysregulation of the immune and inflammatory systems due to CTX ingestion has been reported in the mouse in vivo and in studies involving the whole blood of humans [156].

As hydrophilic compounds, MTXs affect cellular Ca^2+^ homeostasis by mediating Ca^2+^ influx [95]. MTX-mediated Ca^2+^ influx induces numerous cellular responses, such as calcium-dependent depolarization in neuronal cells [157], phosphoinositide breakdown [158], and the contraction of intestinal smooth muscle [159]. Given their potent toxicity, research attention has increasingly focused on the potential medicinal value of MTXs. However, although they represent a unique pharmacological tool for investigating calcium transport, MTXs have not been employed for this purpose owing to the difficulties associated with their purification and artificial synthesis.

## 8. Conclusions and Perspectives

During the past half-century, considerable effort has been dedicated to elucidating *Gambierdiscus* biology, and substantial progress has been made regarding the taxonomy, phylogenetic, geographic distribution, toxin detection method, and toxin biosynthesis of these dinoflagellates. These advances benefit the prevention and management of CFP worldwide. However, owing to their large genome size, unique gene structure, and high gene copy number, little is known about the genome of *Gambierdiscus*. Although a few transcriptome-based studies have been undertaken, this knowledge gap impedes our understanding of *Gambierdiscus* as well as the subsequent efficient prevention and management of CFP. Accordingly, whole-genome sequencing of different *Gambierdiscus* species is urgent and necessary. It will contribute to unveiling the genetic features, evolutionary history, environmental adaptability, and mechanisms of toxin biosynthesis of this genus. The combination of second and third-generation DNA sequencing technologies provides the opportunity to decode the genome of *Gambierdiscus*.

Although various biological and chemical methods have been developed to detect and characterize CTXs and MTXs, fast, simple, specific, and sensitive detection methods are still lacking, primarily owing to the complex structure and diversity of toxin congeners. Additionally, there is an acute lack of purified CTXs and MTXs globally, which impedes the development of toxin detection methods and applications. Thus, there is a need for the isolation and large-scale culture of different *Gambierdiscus* species with different toxin-producing abilities. These will provide sufficient amounts of purified toxins for developing specific detection methods as well as for other applications such as toxicology and pharmacology.

Finally, the responses and adaption of *Gambierdiscus* to the intensification of anthropogenic activities and global warming should be taken into consideration in future studies on the toxicity of these organisms to human beings. Laboratory studies of different *Gambierdiscus* species under various environmental conditions are needed, especially those relating to temperature, irradiance, and nutrients. Meanwhile, a field survey of the geographic distribution and toxicity of *Gambierdiscus* species in all the oceans of the world will aid our understanding of the responses and adaption of *Gambierdiscus* to environmental changes caused by the above-mentioned stresses.

## Figures and Tables

**Figure 1 toxins-14-00485-f001:**
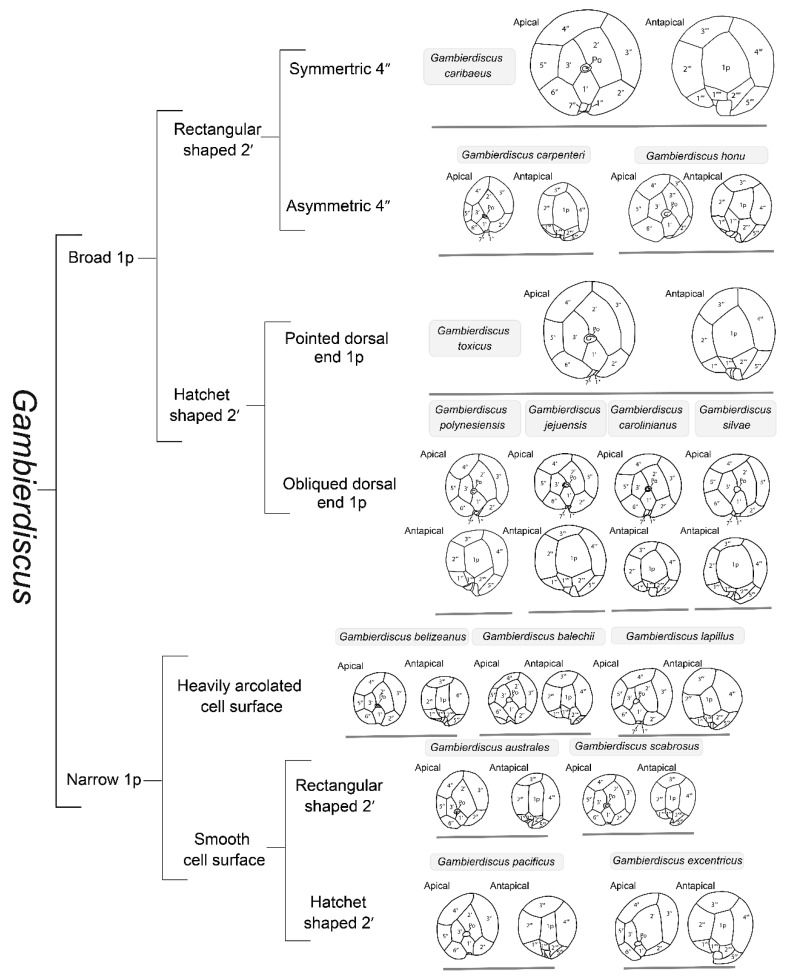
Schematic diagram of the identification of different species of *Gambierdiscus* according to their morphology. Because of the wide variability in *Gambierdiscus* cell size, the size of the line drawings does not reflect the true differences in cell sizes. The line drawings in the figure are modified from: *G. jejuensis* [33], *G. honu* [34], *G. excentricus* [35], *G. toxicus* [30], *G. australes* [30], *G. belizeanus* [30], *G. pacificus* [30], *G. caribaeus* [30], *G. carolinianus* [30], *G. carpenteri* [30], *G. polynesiensis* [30], *G. silvae* [36], *G. cheloniae* [37], *G. balechii* [38], *G. lapillus* [39], and *G. scabrosus* [40].

**Figure 2 toxins-14-00485-f002:**
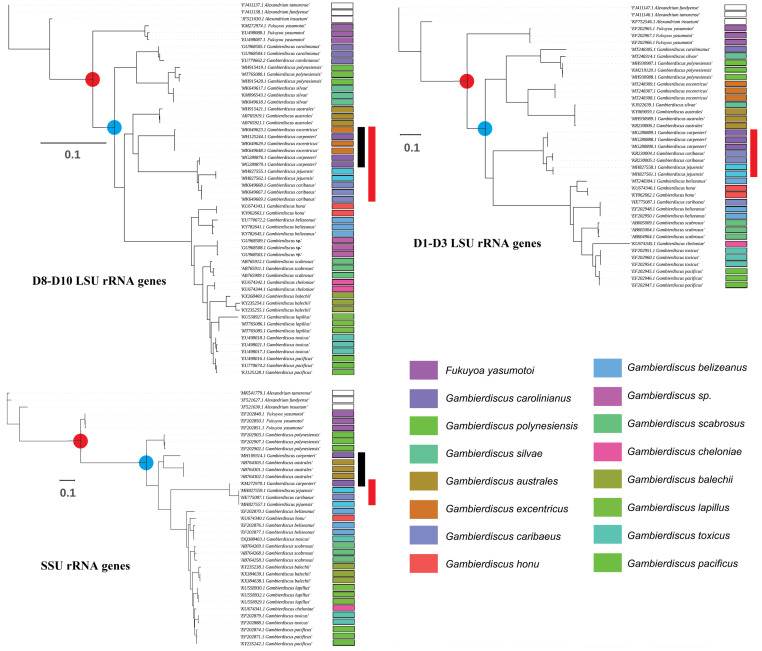
Phylogenetic trees of *Gambierdiscus*. Maximum likelihood phylogenetic trees were constructed based on the LSU D8–D10 rRNA, LSU D1–D3 rRNA, and SSU rRNA genes of *Gambierdiscus*. Different colors are used to label different species, the branching points of *Gambierdiscus* and *Fukuyoa* are marked with red plots, and the first branching point of *Gambierdiscus* is marked with blue plots (the distance between the red and blue plots in the SSU tree is greater than that in the LSU trees).

**Figure 3 toxins-14-00485-f003:**
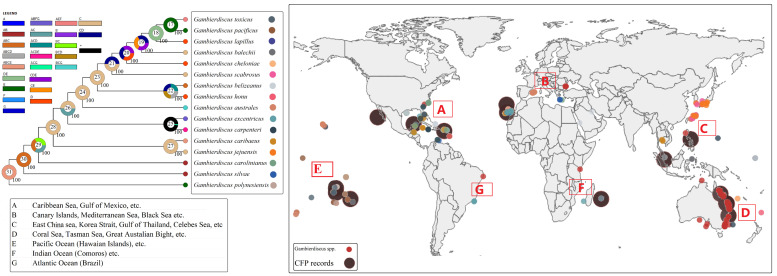
Global distribution of ciguatera food poisoning (CFP) records and *Gambierdiscus* spp. The locations where *Gambierdiscus* are present are classified into six regions (**A**–**G**), and the pie charts in the phylogenetic tree show the probability of the locations at each node. The colors of the point on the right side of the phylogenetic tree are used to distinguish different *Gambierdiscus* species in the global ocean. Distribution information is obtained from the Ocean Biodiversity Information System and the IOC Harmful Algal Bloom Programme (Searched on 23 August 2021) [43,53,54].

**Figure 4 toxins-14-00485-f004:**
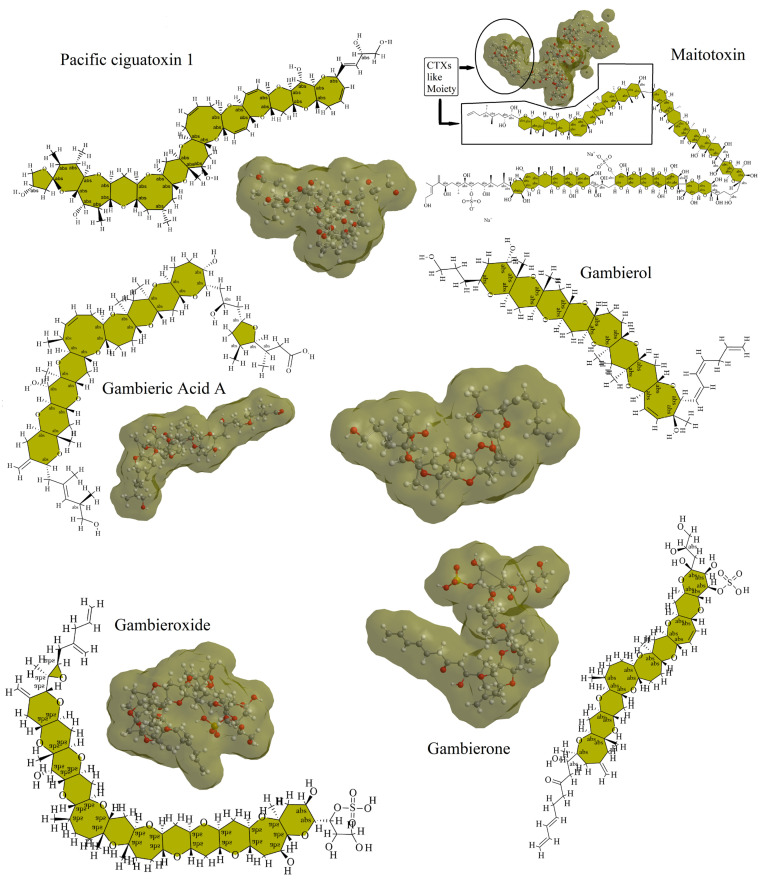
The predicted 2D and 3D structures of Pacific ciguatoxin 1, maitotoxin, and other products of *Gambierdiscus* spp. The framed part indicates the CTX-like moiety, which is the hydrophobic part of the molecule.

**Table 1 toxins-14-00485-t001:** Reported polyether compounds in *Gambierdiscus*.

Species	Ciguatoxins (CTXs)	Maitotoxins (MTXs)	Others	References
*Gambierdiscus australes*	CTX1B, P-CTX-3C	MTX, MTX-3	P-Gambierone analogue, putative gambieroxide	[7,82,83]
*Gambierdiscus balechii*			gambierone	[84]
*Gambierdiscus belizeanus*		MTX-3		[85]
*Gambierdiscus cheloniae*		MTX-3	gambierone	[6,86]
*Gambierdiscus excentricus*		MTX-4		[83]
*Gambierdiscus honu*		MTX-3		[6,34]
*Gambierdiscus pacificus*	51-hydroxyCTX-3C, 2,3-dihydroxyCTX-3C	MTX-3		[6,87]
*Gambierdiscus polynesiensis*	P-CTX-4A, P-CTX-4B, P-CTX-3C, M-seco-CTX-3C, 49-epiCTX-3C	MTX-1, MTX-3		[31]
*Gambierdiscus toxicus*	P-CTX-3C, 2,3-dihydroxy P-CTX-3C, P-CTX-4A/B		Gambieric acids, gambierol, gambieroxide	[88,89,90]

**Table 2 toxins-14-00485-t002:** Examples of CTX detection methods and their characteristics.

Detection Methods	Advantages	Shortcomings	Commercialized Kits
Mouse bioassay	Easy to use	Expensive, lacks specificity, not sensitive enough, and ethical concerns	
Mouse neuroblastoma cell-based assay (CBA-N2a)	Automatable	Expensive, time-consuming, requires specific instruments, and lacks specificity [132]	
Radioimmunoassay	Sensitive	Expensive, time-consuming, and requires specific instruments	
Fluorescent receptor binding assay	Fast	Detection limit is higher than for the CBA-N2a [126]	SeaTox^®^ F-RBA [126]
Enzyme immunoassay	Easy to use	Cross-reaction with other polyether compounds [25]	Ciguatect™ [127]
Antibody-based immunoassays	Sensitive, field application	Cross-reaction with okadaic acid [121]	
Membrane immunobead assay	Specificity	Variation in signal strength [128]	Cigua Check^®^ [128]
Enzyme-linked immunosorbent assay (ELISA)	Sensitive, low detection limit	Need laboratory conditions, require anti-CTX antibodies	CTX-ELISATM 1B [133]
Capillary electrophoresis-based immunoassay	Faster than ELISA	Need laboratory conditions, require anti-CTX antibodies	
Electrochemical immunosensors	Low cost, integrable		
LC–MS/MS	Sensitive, selective	Lack of reference toxins, cannot be used in the field	

## Data Availability

The code and data for this study are available at: https://github.com/yayan-web, accessed on 1 May 2022.

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
