# Peer review of "Gambierdiscus and Its Associated Toxins: A Minireview"

_toxins, 2022, doi:10.3390/toxins14070485_

Round 1
Reviewer 1 Report
General comments
This paper attempts to review several aspects related to the taxonomy, phylogenetics, geographic distribution, environmental regulation, detection method, toxin biosynthesis, and pharmacology and toxicology of Gambierdiscus species.
Being a review, the paper seems short, perhaps better call it a minireview?
The paper is clear (except some points), and it touched a wide range of aspects. My mayor criticisms are the following:
In my opinion this review should considered both Gambierdiscus and Fukuyoa genera together, as
- both are involved in the CFP
- they are closely related genera (species of the genus Fukuyoa have been included into Gambierdiscus until 2015)
- all the scientific literature before the institution of the genus Fukuyoa by Gomez is referred to genus Gambierdiscus sensu lato (i.e. including the globular species now included in Fukuyoa)
- In any case, in a review regarding Gambierdiscus, a detailed explanation regarding the separation between these two genera is expected
In the chapter of biogeographical distribution, Authors completely neglected the Mediterranean Sea, where Gambierdiscus occurs demonstrating that its areal is expanding even in temperate areas.
Among the environmental parameters, the role of nutrients and depth are completely missing and should be commented and discussed. Moreover author should consider that the environmental parameters can affect in different ways the growth of Gambierdiscus and its toxin production (see below).
I suggested to consider some more refs (in which I am not involved) to enlarge the discussion.
Specific comments
L 22. The term ‘epiphytes’ means ‘organisms living on plants’. As Gambierdiscus lives not only on macrophytes, but (as authors correctly wrote) on a variety of benthic substrata (macrophytes, corals, sand grains etc.), I recommend writing as follows: “Gambierdiscus species are autotrophic benthic microalgae living on macrophytes, corals, and sand grains”.
L 23. I don’t think that the ref no. 3 is appropriate in this point.
L 46. The genus Fukuyoa is not simply ‘another benthic alga’, but is a genus closely related to Gambierdiscus (see general comments) and here a more detailed presentation about the differences between these two genera is recommended. Consider that Fukuyoa has been included in the genus Gambierdiscus until 2015.
L 47. The sentence “but relatively few overviews on Gambierdiscus itself have been reported” suggests that separate overviews could be advantageous. Which is the advantage to make separate overviews?
L 51. Add ‘environmental requirements’ to the list of the topics. Specify if the ‘detection method’ you will address is about the toxin or about the microalga.
L 57. Write ‘seagrasses’. I suggest adding ‘and to other substrata’ after 'seagrasses'.
L 63. As in line 61 you wrote ‘armored dinoflagellate’ that means ‘thecate dinoflagellate’, avoid here to repeat ‘are covered by a theca’. The plate formula is Po, 3’, 7’’, 6c, 6-8s, 5’’’, 1p, 2’’’’ as can be seen by your Figure 1, and following both Litaker et al 2009 (that is the reference you cited) and Hoppenrath et al 2014 (that I suggest citing, see below suggestions about refs). I know that some authors like Fraga et al used another criterion for plate formula, but I think it is better to follow the recommendation by Hoppenrath et al 2014 that should be cited immediately after the plate formula, Use the character ‘ and repeat it 1, 2, 3 or 4 times. Don’t use the character “.
L 68-72. In the list:
- holmesii is missing
- lewisi should be corrected as G. lewisii
- pacificus is repeated 2 times
- Refs 34 and 35 are not appropriate for the species list. Instead, cite the Algaebase website that is the updated references for all algal nomenclature, as follows: Guiry, M.D. & Guiry, G.M. 2022. AlgaeBase. World-wide electronic publication, National University of Ireland, Galway. https://www.algaebase.org; searched on the (put the day in which you performed the searching).
Figure 1. Correct the names G. caribaeus in the label inside the figure and G. carpenteri in the legend.
Figure 2. Correct ‘Fukuyoa’ in the legend.
L 81-87. It not expected that the morphological approach would reflect the evolutionary relationships. On the contrary it is expected that it allows species identification. Therefore write as follows. “Furthermore, the morphological approach alone does not properly allow a correct identification at the species level and should be coupled with molecular analyses. The sequencing of ribosomal (r)RNA-encoding DNA, including SSU rRNA, D1–D3 LSU rRNA, and 85 D8–D10 rRNA genes, has been employed for the identification of Gambierdiscus species since the 1990s [44–46]. Gambierdiscus species show similar phylogenetic relationships in phylogenetic trees etc etc”
L 106-172. In this Chapter, the Mediterranean Sea has never been mentioned. Gambierdiscus has been reported not only at tropical and subtropical latitudes but even in temperate areas. Authors mention ‘cold waters’ (L 111). I suggest changing ‘cold waters’ with ‘temperate waters’. Add the Mediterranean Sea in the mentioned areas with appropriate references. I suggest changing the title of this Chapter as ‘Geographic distribution and environmental requirements’.
L 113-118. Rephrase this part, as it is not clear. There are short sentences that are not fluent and appear unrelated each other. I argue that you would express the fact that global warming is expected to increase the seawater temperature, and this would facilitate the expansion of Gambierdiscus distribution. Note that the role of temperature will be already discussed below at lines 141 to 156. Therefore, to avoid confusion I suggest discussing first the present biogeographical distribution then the environmental requirements and after the expected expansion of areal distribution due to global warming, coral reef degradation etc. In alternative, you can separate this Chapter in two chapters. 1 Biogeographical distribution and 2 Environmental requirements’.
L 118-125. ‘low-temperature’ is unclear. Specify the values.
L 141-Regarding the environmental requirements the nutrient role (nitrogen, phosphorous) is completely missing. Does Gambierdiscus grow better in high or low nutrient conditions? Does the eutrophication facilitate or not the expansion of Gambierdiscus? Mention even the depth distribution (you can relate the depth to the irradiance).
The effects of environmental factors should be considered in terms both as effects on growth and of toxin production, given that such effecs can differ.
L 161-162. The salinity measure is adimensional and the udm psu is no longer in use. Remove.
L 162. Change ‘more extreme salinity environments’ with ‘a wider salinity range’.
L 162-163. This sentence seems unnecessary. Consider that heterotrophic diatoms are very rare (maybe only Nitzschia alba does exist). On the contrary heterotrophic dinoflagellates are very common both in phytoplankton and in microphytobenthos (a half of dinoflagellate species lose their plastids and are heterotrophic).
L 187. “it was since found to be produced instead by…” Sentence unclear. Maybe ‘since’ could be removed?
L 246. Associated inherent flaws? Unclear.
L 254. Add references after ‘commercialized’. The citation of Table 1 seems not appropriate as the fact the commercialization of assays is not reported.
L 257. ‘toxin references’ .. do you mean ‘standard’?
L 259. Add a comma after ‘years’
L 293-297. The mentioned ‘recent comparative study …. etc ‘ lacks the reference. Note that the species Ceratium balechii has been renamed as Tripos dens. If in the original paper it is reported as Ceratiuim balechii, write as follows: ‘…. strains of Tripos dens (formerly Ceratium balechii) ….”
References
A number of refs are wrongly written, and the whole list should be checked. In all the references’ titles, all the words are capitalised. This is not in line with the editorial requirements and is boring for the reader.
- All the species names should be written in italics.
- Genus name should be capitalized, while and species name should be uncapitalized
- nov. should be uncapitalised
- sp. and spp. should be uncapitalised and not Italic
L 376. inclinatum (uncapitalized)
L 366. Dinoflagellate
L 408 Gambierdiscus in italic
L 427. The editor of the volume is Subba Rao D.V, Ed.
L 435. gen. et sp. nov. (uncapitalized)
L 454 Gambierdiscus in italic
L 510. Ref. Acosta: Incomplete list of Authors, Incomplete doi, Page numbers missing
L 546. Cabra matta gen. nov., sp. nov. (only matta should be uncapitalised)
L 553. Latus (uncapitalized)
L 582. Gambierdiscus pacificus (italic and species name uncapitalized)
L 718. Gambierdiscus in italic
L 720 Gambierdiscus polynesiensis. Italic and species uncapitalized
L 723. Polysiphonia fastigiata. Italic and species uncapitalized
Ref. n. 28 seems incomplete
Refs n. 33 and 48 are identical
Some refs to be cited
M.L. Chateau-Degat, M. Chinain, N. Cerf, et al. Seawater temperature, Gambierdiscus spp. variability and incidence of ciguatera poisoning in French Polynesia Harmful Algae, 4 (2005), pp. 1053-1062
Mar Drugs. 2021 Aug 15;19(8): 460.doi: 10.3390/md19080460.
Hoppenrath M., Murray S.A., Chomérat N., Horiguchi T., 2014. Marine benthic dinoflagellates - unveiling their worldwide biodiversity. Kleine Senckenberg-Reihe, Band 54, Schweizerbart, Stuttgart, Germany, 276 pp.
RW Litaker, MW Vandersea, MA Faust, SR Kibler etc. 2010 - Global distribution of ciguatera causing dinoflagellates in the genus Gambierdiscus- Toxicon
Loeffler, C.R., Richlen, M.L., Brandt, M.E., Smith, T.B., 2015. Effects of grazing, nutrients, and depth on the ciguatera-causing dinoflagellate Gambierdiscus in the US Virgin Islands. Mar. Ecol. Prog. Ser. 531, 91–104.
Longo, S.; Sibat, M.; Darius, H.T.; Hess, P.; Chinain, M. Effects of pH and Nutrients (Nitrogen) on Growth and Toxin Profile of the Ciguatera-Causing Dinoflagellate Gambierdiscus polynesiensis (Dinophyceae). Toxins 2020, 12, 767. https://doi.org/10.3390/toxins12120767
F Rodríguez, S Fraga, I Ramilo, P Rial, RI Figueroa… Canary Islands (NE Atlantic) as a biodiversity 'hotspot'of Gambierdiscus: Implications for future trends of ciguatera in the area- Harmful Algae, 2017 -
À Tudó, A Toldrà, M Rey, I Todolí, KB Andree… - Gambierdiscus and Fukuyoa as potential indicators of ciguatera risk in the Balearic Islands- Harmful Algae, 2020
Reviewer 2 Report
The manuscript entitled "Gambierdiscus and its associated toxins: An overview" addresses a relevant and appropriate topic for this journal.
Overall this manuscript is well structured and well-founded. A few corrections should be introduced to improve the manuscript.
Corrections needed:
line 55 - Gambierdiscus species are autotrophic, benthic dinoflagellates (phylum Miozoa, class Dinophyceae) with yellow to brown
line 102 - are used to label different species, the branching points of Gambierdiscus and Fukuyoa are marked with red plots, and the
line 163 - ... benthic diatoms (e.g., Nitzschia sp.[78]) ...
line 294 - CTX-producing strain of Tripos dens (formerly Ceratium balechii) (Miozoa) identified 52 PKS genes that were upregulated
line 297 - and potentially also CTX synthesis in T. dens. ...
line 321 - ... The major symptoms of ciguatoxin poisoning occur within 1-3 h of
line 322 - ... as vomiting, diarrhea, numbness in the extremities, mouth and lips, reversal of hot and cold sensations, and muscle and joint aches.
Reviewer 3 Report
The review is devoted to the problem of ciguatoxins (CTXs), maitotoxins (MTXs) and related toxins, which are produced by unicellular marine dinoflagellates belonging to the Gambierdiscus genus. Poisoning occurs when eating CTX-contaminated fish or shellfish; the number of patients is estimated at 50-500 thousand per year. The anthropogenic factor and climate warming lead to the spread of dinoflagullates and an increase in the number of cases. The review provides data on almost all aspects of the problem: taxonomy and phylogenetics of Gambierdiscus, geographic distribution and environmental adaptability, methods for determining these toxins, Gambierdiscus-associated toxins, CTXs and MTXs biosynthesis, toxicology and pharmacology of CTXs.
It is concluded that whole-genome sequencing of different Gambierdiscus species is urgent and necessary; fast, simple, specific, and sensitive detection methods for CTXs and MTXs are still lacking; there is a need for the isolation and large-scale culture of different Gambierdiscus species with different toxin-producing abilities to provide sufficient amounts of purified toxins for developing specific detection methods as well as for other applications such as toxicology and pharmacology.
A very high quality review, excellently written and designed, fully revealing all aspects of the problem of ciguatera fish poisoning.
Author Response
Dear reviewer, thank you so much for your review! We deeply appreciate your recognition of our work.
Round 2
Reviewer 1 Report
The paper has been revised following my suggestions and seems improved compared to the previous version. However after the revision I have other suggestions mainly referred to the revised parts.
The references need a careful revision.
Put the verbs in the past form throughout the text. Revise English.
Minor comments
L 20. Key Contribution. Change ‘environmental regulation’ with ‘role of environmental factors’.
L 26. Remove ‘generally’.
L 43. Put a comma after ‘method’, and remove ‘toxin’ before biosynthesis’.
L 52. Change ‘environmental regulation’ with ‘role of environmental factors’.
L 56-64. Rephrase all as follows: Gambierdiscus (Gonyaulacales, Dinophyceae) species are armored, benthic dinoflagellates predominantly living in coral reef ecosystems attached through mucous filaments to the surfaces of macroalgae, seagrasses, and other substrata [3,27]. The morphology of Gambierdiscus has been extensively studied since 1978. Cells are large sized (diameter 42 to 140 μm) [28], with a strong anteroposterior compression and an ascending cingulum with a recurved distal end and contains several yellow to brown chloroplasts [26]. Gambierdiscus species are traditionally identified based on subtle differences in their thecal plate morphology as observed under light microscopy and scanning electron microscopy [26,29]. According to the Kofoidian nomenclature ….. etc.
L 68. The citation n. 11 is not appropriate in this context. Remove.
L 73. G. holmesii is repeated 2 times, while G. pacificus is lacking. Delete holmesii at line 73 and insert pacificus.
L 115. Change environmental regulation’ with ‘role of environmental factors’
L 125-137. This paragraph regarding biogeography appears not so fluent after revision. Rephrase as follows: “In a phylogenetic analysis-based study, Litaker et al. (2010) reported that five Gambierdiscus species are endemic to the Atlantic (including the Caribbean/West Indies and Gulf of Mexico), five are endemic to the tropical Pacific, and that two (G. carpenteri and G. caribaeus) are globally distributed. However, G. belizeanus, an Atlantic species following Litaker et al. [55], was later reported in the Central Pacific by Xu et al. [56], suggesting that some Gambierdiscus species might have been transferred via modern shipping activities [59]. Rodríguez et al. suggested that the Canary Islands (North-East Atlantic) could represent ancient settlement sites for Gambierdiscus as suggested by the high species diversity in the area [60], however, there is still not enough evidence to prove this hypothesis. The dispersal-vicariance analysis performed in this study using RASP (Figure 3) [57] showed that some widely distributed species, such as G. carpenteri and G. caribaeus, were scattered in different clades of the tree.”
L 164. Reference n. 3 is not appropriate here. Remove.
L 174. Put a paragraph separation when you change the environmental factor.
L 177. Change ‘abundant cells’ with ‘higher abundances of Gambierdiscus species are observed in shallower than in deeper waters”.
L 184-186. Enlarge the part of grazing. Being a review, the reader expects more details. Which are the main consumers of Gambierdiscus? How are the toxins transferred along the trophic chain? Explain that the grazers, while removing Gambierdiscus affecting their abundances, become at the same time vectors for Ciguatera. Explain in which fish tissues the toxins accumulate.
L 285. All the 1st paragraph does not report in which environmental conditions the toxin production (intended as toxin content per cell) increases. I invite Authors to enlarge this part
L 286. Start with ‘Several studies …”.
L 287. ‘carpenteri’
L 291. … in a field study
References
I noted that in a number of references the doi number is very short and does not work. Is it correct?
Check the references one by one!. Some are incomplete with volume and/or pages/article numbers missing, some have authors's names with wrong accent marking, in some the Latin names are not in italic.
Just as examples (I have no intention to do this checking for all
1. Check the spelling Hoppenrath. Write this ref as follows: Hoppenrath M., Murray S.A., Chomérat N., Horiguchi T. Marine benthic dinoflagellates - unveiling their worldwide biodiversity. Kleine Senckenberg-Reihe, Band 54, Schweizerbart, Stuttgart, Germany, 2014 ISBN 978-3-510-61402-8
6. The Journal of this ref. is Mar. Drugs and not ‘Mar.An Updated Review of Ciguatera Fish Poisoning: Clinical, drugs’.
11. Gambierdiscus in italic.
15. The article number is missing. Write 9(10): 2291, doi:…..
16. Check the accent mark and correct as follows: Tudó, À.; Gaiani, G.; Rey Varela, M.; Tsumuraya, T.; Andree, K.B.; Fernández-Tejedor, M.; Campàs, M.; Diogène, J.
20. Page (or article) number is missing.
21. Page (or article) number is missing
34. sp.
44. Accent marking.
60. Figueroa, R.I.; Riobó, P.
to be continued....
Author Response
Response to reviewer’s comments
Question 1: L 20. Key Contribution. Change ‘environmental regulation’ with ‘role of environmental factors’.
Response: Corrected and please see line 20.
Question 2: L 26. Remove ‘generally’.
Response: Corrected and please see line 26.
Question 3: L 43. Put a comma after ‘method’, and remove ‘toxin’ before biosynthesis’.
Response: Corrected and please see line 43.
Question 4: L 56-64. Rephrase all as follows: Gambierdiscus (Gonyaulacales, Dinophyceae) species are armored, benthic dinoflagellates predominantly living in coral reef ecosystems attached through mucous filaments to the surfaces of macroalgae, seagrasses, and other substrata [3,27]. The morphology of Gambierdiscus has been extensively studied since 1978. Cells are large sized (diameter 42 to 140 μm) [28], with a strong anteroposterior compression and an ascending cingulum with a recurved distal end and contains several yellow to brown chloroplasts [26]. Gambierdiscus species are traditionally identified based on subtle differences in their thecal plate morphology as observed under light microscopy and scanning electron microscopy [26,29]. According to the Kofoidian nomenclature ….. etc.
Response: Thanks for the reviewer’s suggestion. Corrected and please see L 56-64.
Question 5: L 68. The citation n. 11 is not appropriate in this context. Remove.
Response: Corrected and the citation n. 11 was moved to L36.
Question 6: L 73. G. holmesii is repeated 2 times, while G. pacificus is lacking. Delete holmesii at line 73 and insert pacificus.
Response: Corrected and please see L71.
Question 7: L 115. Change environmental regulation’ with ‘role of environmental factors’
Response: Corrected and please see L114.
Question 8: L 125-137. This paragraph regarding biogeography appears not so fluent after revision. Rephrase as follows: “In a phylogenetic analysis-based study, Litaker et al. (2010) reported that five Gambierdiscus species are endemic to the Atlantic (including the Caribbean/West Indies and Gulf of Mexico), five are endemic to the tropical Pacific, and that two (G. carpenteri and G. caribaeus) are globally distributed. However, G. belizeanus, an Atlantic species following Litaker et al. [55], was later reported in the Central Pacific by Xu et al. [56], suggesting that some Gambierdiscus species might have been transferred via modern shipping activities [59]. Rodríguez et al. suggested that the Canary Islands (North-East Atlantic) could represent ancient settlement sites for Gambierdiscus as suggested by the high species diversity in the area [60], however, there is still not enough evidence to prove this hypothesis. The dispersal-vicariance analysis performed in this study using RASP (Figure 3) [57] showed that some widely distributed species, such as G. carpenteri and G. caribaeus, were scattered in different clades of the tree.”
Response: Thanks for the reviewer’s suggestion. Corrected and please see L 124-135.
Question 9: L 164. Reference n. 3 is not appropriate here. Remove.
Response: Corrected and please see L162.
Question 10: L 174. Put a paragraph separation when you change the environmental factor.
Response: Corrected and please see L 174 and L181.
Question 11: L 177. Change ‘abundant cells’ with ‘higher abundances of Gambierdiscus species are observed in shallower than in deeper waters”.
Response: Corrected and please see L175-176.
Question 12: L 184-186. Enlarge the part of grazing. Being a review, the reader expects more details. Which are the main consumers of Gambierdiscus? How are the toxins transferred along the trophic chain? Explain that the grazers, while removing Gambierdiscus affecting their abundances, become at the same time vectors for Ciguatera. Explain in which fish tissues the toxins accumulate.
Response: Thanks for the reviewer’s comments and suggestions. We added some information and please see L181-194.
Question 13: L 285. All the 1st paragraph does not report in which environmental conditions the toxin production (intended as toxin content per cell) increases. I invite Authors to enlarge this part
Response: Thanks for the reviewer’s comments and suggestions. We added some information and please see L 290-294.
Question 14: L 286. Start with ‘Several studies …”.
Response: Corrected and please see L 291.
Question 15: L 287. ‘carpenteri’
Response: Corrected and please see L 293.
Question 16:. L 291 … in a field study
Response: Corrected and please see L 297.
Question 17: I noted that in a number of references the doi number is very short and does not work. Is it correct?
Response: Thanks for the reviewer’s comment. In our previous version, we used the shortDOI Service which produced shortened DOI® names, with the form 10/abcde, as the aliases for the existing DOI names which are usually long. In order to avoid misunderstanding, we replaced the short DOIs with the full DOIs of the references.
Question 18: Check the references one by one!. Some are incomplete with volume and/or pages/article numbers missing, some have authors's names with wrong accent marking, in some the Latin names are not in italic.
Response: Thanks for the reviewer’s comment. We doubled checked the references and corrected those errors in the references, please see refs. 1, 8, 10, 13, 16, 18, 22, 24, 45, 51, 58, 73, 83, 85, 99, 113, 128, 143, 149.
Question 19: 1. Check the spelling Hoppenrath. Write this ref as follows: Hoppenrath M., Murray S.A., Chomérat N., Horiguchi T. Marine benthic dinoflagellates - unveiling their worldwide biodiversity. Kleine Senckenberg-Reihe, Band 54, Schweizerbart, Stuttgart, Germany, 2014 ISBN 978-3-510-61402-8
Response: Corrected and please see L 414-415.
Question 20: 6. The Journal of this ref. is Mar. Drugs and not ‘Mar.An Updated Review of Ciguatera Fish Poisoning: Clinical, drugs’.
Response: Corrected and please see L 427.
Question 21: 11. Gambierdiscus in italic.
Response: Corrected and please see L 441.
Question 22: 15. The article number is missing. Write 9(10): 2291, doi:….
Response: Corrected and please see L 451.
Question 23: 16. Check the accent mark and correct as follows: Tudó, À.; Gaiani, G.; Rey Varela, M.; Tsumuraya, T.; Andree, K.B.; Fernández-Tejedor, M.; Campàs, M.; Diogène, J.
Response: Corrected and please see L 452.
Question 24: 20. Page (or article) number is missing.
Response: Corrected and please see L 464.
Question 25: 21. Page (or article) number is missing.
Response: Corrected and please see L 467.
Question 26: 34. sp.
Response: Corrected and please see L 500.
Question 27: 44. Accent marking.
Response: Corrected and please see L 528.
Question 28: 60. Figueroa, R.I.; Riobó, P.
Response: Corrected and please see L 562.